# Healthy Lifestyle Score and Incidence of Glaucoma: The Sun Project

**DOI:** 10.3390/nu14040779

**Published:** 2022-02-12

**Authors:** Javier Moreno-Montañés, Elsa Gándara, Itziar Gutierrez-Ruiz, Laura Moreno-Galarraga, Miguel Ruiz-Canela, Maira Bes-Rastrollo, Miguel Ángel Martínez-González, Alejandro Fernandez-Montero

**Affiliations:** 1Department of Ophthalmology, Clínica Universidad de Navarra, 31008 Pamplona, Spain; jmoreno@unav.es (J.M.-M.); egandararod@unav.es (E.G.); gutierrez.itziar@gmail.com (I.G.-R.); 2IdiSNA (Instituto de Investigación Sanitaria de Navarra), 31008 Pamplona, Spain; lauramoreno11@yahoo.es (L.M.-G.); mcanela@unav.es (M.R.-C.); mbes@unav.es (M.B.-R.); mamartinez@unav.es (M.Á.M.-G.); 3Department of Preventive Medicine and Public Health, School of Medicine, University of Navarra, 31008 Pamplona, Spain; 4Department of Pediatrics, Complejo Hospitalario de Navarra, Servicio Navarro de Salud, 31008 Pamplona, Spain; 5CIBER Fisiopatología de la Obesidad y Nutrición (CIBER Obn), Instituto de Salud Carlos III, 28029 Madrid, Spain; 6Department of Nutrition, Harvard TH Chan School of Public Health, Boston, MA 02115, USA; 7Department of Occupational Medicine, University of Navarra, 31008 Pamplona, Spain

**Keywords:** mediterranean diet, mediterranean lifestyle, glaucoma, cohort

## Abstract

Background: The relationship between modifiable risk factors, such as diet and lifestyle, and glaucoma remains controversial. We analyse the effect of the Mediterranean lifestyle (ML) on glaucoma incidence in the “Seguimiento Universidad de Navarra” (SUN) Project. Methods: The SUN Healthy Lifestyle Score (SHLS) includes 10 healthy habits: never having smoked, moderate to high physical activity, Mediterranean diet adherence, moderate alcohol consumption, low television exposure, no binge drinking, short afternoon napping, meeting up with friends, working at least 40 h/wk, and low body mass index. The information was collected biennially through self-reported questionnaires. The relationship between new glaucoma cases and the SHLS was assessed by Cox regression using hazard ratios. Crude, multi-adjusted, and sensitivity analyses were performed. Results: During a median of 12 years of follow-up, 261 (1.42%) new cases of glaucoma were identified among 18,420 participants. After adjusting for potential confounders, participants in the healthiest SHLS category showed a significantly reduced risk of glaucoma compared to those in the lowest SHLS category (adjusted HR = 0.51, 95% CI = 0.28–0.93). For each point added to the SHLS, the risk of glaucoma relatively dropped 5%. Conclusions: Higher adherence to a ML, measured by the SHLS, was significantly associated with a lower risk of developing glaucoma. Based on our study, the ML is a protective factor for glaucoma incidence.

## 1. Introduction

Glaucoma is one of the most common eye diseases and includes a group of disorders characterized by the progressive damage of the optic nerve associated with loss of the visual field [1]. It is considered the second leading cause of blindness worldwide, and a major public health problem [2]. According to the European Glaucoma Society, the number of people with glaucoma was 76 million in 2020 and is estimated to increase to 112 million in 2040. In addition to early detection of the disease and early reduction in intraocular pressure (IOP), several approaches have been made to search for risk factors in order to reduce the incidence of glaucoma. Some of these risk factors cannot be modified, such as age [3], race [4], or myopia [5], among others. However, the intervention on modifiable risk factors, such as those related to lifestyle or diet, may reduce the incidence and progression of glaucoma. In this regard, some of our previous studies obtained that smoking cessation significantly reduced the incidence of glaucoma [6]. Additionally, in a previous study we found that high omega 3:6 ratio intake was associated with a significantly higher risk of glaucoma incidence [7]. Kang et al., in a prospective analysis from the Nurses’ Health Study and the Health Professionals Follow-up Study, found that higher dietary nitrate and green leafy vegetable intake was associated with a lower glaucoma risk [8]. Despite these results, the relationship between lifestyle and glaucoma incidence remains unclear and previous studies even reported contradictory results, therefore this association requires further analysis, especially among larger cohort groups and with longer follow-up periods.

Two large studies, in which members of our group collaborated, demonstrated the efficacy of the Mediterranean diet (MD) in preventing cardiovascular diseases and reducing mortality. Thus, the “Seguimiento Universidad de Navarra” (University of Navarra Follow-up) (SUN study), in a cohort of 20,127 participants, found that poor adherence to the Mediterranean diet and low physical activity levels accounted for almost 30% of all-cause mortality [9]. Another multicentre study, the “Prevencion con Dieta Mediterránea” (PREDIMED study), which included 7447 participants at high cardiovascular risk followed up for 5 years, demonstrated that the incidence of major cardiovascular events was lower among those assigned to a Mediterranean diet supplemented with extra-virgin olive oil or nuts than among those assigned to a reduced-fat diet [10]. These studies suggest the health benefits of the Mediterranean diet. However, the Mediterranean diet is only a part of the Mediterranean culture and lifestyle (ML) [11]. To evaluate adherence to the ML, we created the SUN Healthy Lifestyle Score (SHLS) which included 10 Mediterranean lifestyle habits. In previous studies, we found that a high SHLS score was associated with a lower risk of developing primary cardiovascular disease events [12] and metabolic syndrome [13]. Our goal in this study is to analyse the association between the ML and glaucoma incidence using the SHLS, in order to find new modifiable risk factors and potential lifestyle changes useful to reduce glaucoma incidence. To our knowledge, no previous studies have been published analysing the association between lifestyle scores and glaucoma incidence.

## 2. Material and Methods

### 2.1. Study Design

The SUN Project is a dynamic prospective cohort study formed of university graduates. It started in Spain in December 1999 and the recruitment of new participants is permanently open. The participants’ information is collected biennially through mailed or electronically mailed self-reported questionnaires. Upon completion of the first questionnaire (Q-0), which includes a total of 554 items used as baseline information, the participants receive follow-up questionnaires biennially. These follow-up questionnaires contain questions to evaluate changes in lifestyle and health-related behaviours, anthropometric measures, incident diseases, and medical conditions. The study methods have been previously published [14].

### 2.2. Study Participants

We preselected a total of 22,475 subjects who had responded to the baseline questionnaire Q-0 before September 2015. We used the information collected after the first 2 years of follow-up (Q2), and after every 2 subsequent years (Q4, Q6, Q8, Q10, Q12, and Q14). The participants were followed up until the diagnosis of glaucoma or until the last follow-up questionnaire available at the time of these analyses (Q16), which collected information after 16 years of follow-up. The data analysis excluded 2071 individuals without follow-up questionnaires (90.8% retention rate). Participants who reported extremely low or high total energy intake [15] (1926) and participants with a previous diagnosis of glaucoma (58) were also excluded. Finally, 18,420 participants were included in the analysis (Figure 1). The study was approved by the Human Research Ethics Committee at the University of Navarra (091/2008) and followed the Declaration of Helsinki Ethical Principles for Medical Research Involving Human Subjects. Voluntary completion of the first questionnaire was considered by this committee to imply informed consent; this handling of consent was approved by the ethics committee.

### 2.3. Exposure Assessment: Healthy Lifestyle Score Variables

Dietary exposure was gathered using a semi-quantitative food frequency questionnaire (FFQ) with consume information on 136 items that had been repeatedly validated in Spain [16]. The Trichopoulou score (score 0–8, excluding alcohol) was used to assess the adherence to the Mediterranean diet [17]. To collect information on alcohol consumption, data were obtained through the FFQ and other additional alcohol-intake related items present in the baseline questionnaire.

The Mediterranean lifestyle was assessed with the SHLS. For the SHLS calculation, one point was given to each participant for each of these 10 habits: never smoking, moderate-to-high physical activity (>20MET-h/wk.), Mediterranean diet (≥4 adherence points), body mass index (BMI) ≤ 22, moderate alcohol consumption (women, 0.1–5.0 g/d; men, 0.1–10.0 g/d; abstainers excluded), low television exposure (<2 h/d), no binge drinking (≤5 alcoholic drinks at any time), taking a short afternoon nap (<30 min/d), regularly meeting up with friends (>1 h/d), and working at least 40 h/wk. The SHLS scale ranged between 0 points (worst lifestyle) and 10 points (best lifestyle).

### 2.4. Outcome Assessment

The diagnosis of glaucoma was self-reported. To evaluate glaucoma incidence, cases of glaucoma detected at the baseline questionnaire (Q-0) were excluded from the analysis. The participants were asked in each follow-up questionnaire whether they had been diagnosed with glaucoma by an ophthalmologist and if so, the date of the diagnosis. Self-reported glaucoma diagnosis in the SUN Project was validated in a subgroup of 150 randomly selected participants. The participants in the validation study signed a written consent giving access to their medical records. The analysis was performed by an experienced ophthalmologist blinded to the participants’ responses. Self-reported diagnosis was confirmed by the medical records according to both structural and functional evidence of glaucomatous optic neuropathy, such as visual field defect consistent with glaucoma or optic disc abnormalities. The validation study between clinical diagnosis and self-reported diagnosis showed almost perfect agreement, according to the Landis and Koch classification [18], with a Kappa value of 0.85 (95% coefficient interval [CI], 0.834–0.872). The sensitivity found was 0.83 and the specificity was 0.99. All cases examined were open-angle glaucoma. When a participant was diagnosed with glaucoma on a questionnaire, he was excluded from the study, so only one case per participant was considered in the analysis.

### 2.5. Ascertainment of Covariates

The baseline questionnaire also gathered information on multiple potential confounding factors, such as socio-demographic characteristics (i.e., sex, age, and educational level), lifestyle and health-related characteristics (i.e., smoking, physical activity, adherence to the Mediterranean diet, total energy intake, consumption of a special diet, caffeine intake, and Omega 3/6 ratio), anthropometric measures (i.e., BMI), and prevalent diseases (i.e., hypertension, cardiovascular disease, cancer, and diabetes).

### 2.6. Statistical Analyses

According to the baseline score obtained with the SHLS, the participants were classified into four groups to ensure an appropriate sample distribution and a sufficient number of incident cases within each category. These four categories were SHLS 0–2, 3–4, 5–6, and 7–10 points. We estimated the hazard ratios (HR) and 95% confidence interval (CI) for glaucoma in each category of the SHLS using the Cox regression model, defining the first category (0–2) as the reference category, and adjusting for multiple potential confounding factors, such as sex, age, calorie intake, caffeine intake, alcohol intake, omega-3/omega-6 ratio, prevalence of cancer, prevalence of hypertension, prevalence of diabetes mellitus type 2, educational level, and special diets. Linear trend tests were calculated by assigning the median score of each category to all participants in that category and treating this variable as continuous.

To analyse the individual contribution of each specific factor of the SHLS score to the risk of glaucoma, Cox regression models were fitted for each of the 10 items of healthy life habits, adjusting for the effect of the rest of the items that constituted the index. The reference category was the absence of the healthy habit (score 0 on the specific item).

Sensitivity analyses were also performed to ensure the robustness of the results in different scenarios. We repeated the analyses stratifying by age (≥50), sex, and smoking.

All *p* values presented are two-tailed; *p* < 0.05 was considered statistically significant. Analyses were performed using STATA/SE version 12.0 (STATA Corp LP, College Station, TX, USA).

## 3. Results

After a median of 12 years of follow-up (mean 10.8 years), we recorded 261 incident cases of glaucoma. The participants’ baseline characteristics according to their classification in the SHLS are shown in Table 1. Compared to the participants at the lowest range of the SHLS (0–3 points), those at the highest category (7–10 points) had a lower BMI, consumed less alcohol, binge drunk less, smoked less, and watched less television. On the other hand, they practiced more physical activity, had higher adherence to the Mediterranean diet, took more weekly naps, met more with friends, and worked more hours per week.

The risk of developing glaucoma according to the Sun Healthy Lifestyle Score (Table 2) was significantly lower for those participants with the highest scores than for those with the lowest SHLS in all three models: the crude-model (adjusted HR = 0.53, 95% CI = 0.3–0.96), the age-sex adjusted model (adjusted HR = 0.54, 95% CI = 0.3–0.97), and the multivariable model (multivariable-adjusted HR = 0.52, 95% CI = 0.29–0.94). For each point added in the SHLS, the risk of glaucoma relatively dropped by 5% (*p* for trends: 0.032).

Figure 2 shows the multivariable-adjusted HRs for the risk of glaucoma across the 10 habits included in the SHLS. Only “never smoking” was associated with low glaucoma incidence (adjusted HR = 0.70, 95% CI = 0.53–0.92) when evaluated individually. However, when combined, the effect of the 10 healthy habits is significantly protective against glaucoma incidence. As shown in Table 3, the contribution of each habit to the SHLS varied from 13.7% (no binge drinking) to 6.6% (BMI).

## 4. Discussion

Several Healthy Lifestyle Scores, combining different variables, have been used to study the association between modifiable lifestyles and systemic diseases. For example, Jiao et al., evaluated five habits in 450,416 participants and found that the combined highest score (five points) was associated with a 58% reduction in the risk of developing pancreatic cancer compared to the lowest combined score [19]. Other authors that combined four lifestyle factors (Mediterranean diet, moderate alcohol use, physical activity, and non-smoking), in individuals aged 70 to 90 years, found that poor adherence to that combined score was associated with a population attributable risk of 60% for all deaths [20]. Another study that monitored 42,847 men using five habits updated through self-reported questionnaires showed that 62% of coronary events in this cohort may have been prevented with better adherence to these five healthy lifestyle practices [21]. Similar results of other scores that included five healthy components (diet, physical activity and sedentary behaviours, smoking, social support and network, and sleep) found that this score was associated with the metabolic syndrome [22]. Recently, another study showed how a favourable lifestyle was associated with a lower risk of dementia, even regardless of the genetic risk [23]. All these results reinforce the extraordinary importance of healthy lifestyle in health prevention.

The Mediterranean lifestyle (that includes not only adherence to the Mediterranean diet, but also other habits, such as high sociability, regular physical activity, moderate drinking, or adequate rest) has already shown its role in the prevention of chronic diseases in hundreds of peer-reviewed scientific journal articles during the last years [24]. The UNESCO recognized the ML as an Intangible Cultural Heritage of Humanity in 2010 [25]. Additionally, the Mediterranean diet has been selected as a model for healthy dietary patterns in the Dietary Guidelines for Americans 2015–2020 of the United States Department of Agriculture [26] and in several clinical guides from both American and European scientific societies [24,27]. In our SHLC, we included 10 ML habits and each of them alone has demonstrated a beneficial effect on health. Some studies demonstrated the relationship between smoking and worse health and chronic diseases [28]. Moderate alcohol consumption, such as red wine or beer, is considered part of the Mediterranean diet, and many epidemiological studies have demonstrated the protective effect of moderate alcohol consumption in coronary heart disease, even more than abstainers [29,30], while binge-drinking (7 drinks 1 day of the week) has been associated with poorer health and strokes [31]. The short nap in the mid-day (“siesta”) has been considered a protective factor of cardiovascular diseases in previous studies conducted in Greece [32] and in Japan [33]. Conviviality or sociability with friends is another known factor for longevity, cardiovascular, and mental health outcomes [28,34]. Watching television (TV) several hours a day is considered a sign of sedentary life and has been implicated in cardiovascular events [35], obesity [36], and mortality [37]. Eating high-calorie foods and sugary drinks while watching a lot of TV may also be related to obesity [36]. A high adherence to the Mediterranean diet has demonstrated a high benefit for the prevention of cardiovascular diseases, diabetes, and various types of cancer [38,39]. The relationship between working hours (>40 h/wk) and health is more controversial [28]. Some studies showed a higher risk of coronary heart diseases in employees working long hours [40], while another study found that longer overtime work was inversely associated with hyperglycaemia or type 2 diabetes in Japanese office workers [41]. In general, we believe that these findings might be affected by the fact that in order to develop or remain in a job that requires long working hours, a better health status is needed [42]. Many reports describe the protective effect of physical activity on preventing heart disease and stroke, high blood pressure, diabetes mellitus, or metabolic syndrome, among others [43]. Finally, a BMI > 30 has been associated with all-cause mortality. In a study with 1.46 million white adults, overweight and obesity were associated with increased all-cause mortality, and all-cause mortality was lowest for those with a BMI between 20.0 and 24.9 [44].

The score we have designed, which includes these 10 items, adequately reflects and captures the complexities of the ML. In isolation, only one habit has been shown to be related to the incidence of glaucoma (smoking), but when analysed together these habits show a significant protective effect. The difference in the effect of each isolated healthy habit, as opposed to the combined effect of several healthy factors, is that “the whole is more than the sum of its parts” (Aristotle). It is logical to deduce that adherence to a number of individually beneficial lifestyle habits would lead to a better synergistic effect [12].

The biological pathway behind this association remains unclear. Some studies suggest that chronic and low-grade inflammation is a key factor in the pathogenesis of glaucoma [45,46]. Interleukin-6 (IL-6) has a major role in the pathology of glaucoma, among other systemic inflammatory diseases [47], and high plasma C-reactive protein (CRP) levels have been associated with normal tension glaucoma [48]. Some of the ML components are associated with better endothelial function, and lower inflammation levels and oxidative stress. Systematic review and meta-analysis showed that a ML significantly improved chronic inflammation markers such as CRP, pro-inflammatory cytokines such as IL-6, and adiponectin (AD), as well as endothelial function parameters, such as intercellular adhesion molecule 1 (ICAM-1), vascular cell adhesion molecule 1 (VCAM-1), and E-Selectin [49]. The PREDIMED study found that high consumption of Mediterranean diet products, such as cereals, fruit, nuts, and virgin olive oil was associated with lower serum concentrations of IL-6, CRP, VCAM-1, and ICAM-1 [50]. These inflammatory markers can be associated with glaucoma incidence explaining the results found.

Another plausible explanation for the association between the ML and glaucoma is through the nitric oxide (NO) pathway. The NO signalling pathway can be favourably manipulated by dietary interventions [51]. The NO serves as a signalling molecule for a wide variety of essential physiological functions, including IOP regulation [51]. In fact, using data from some health professional cohorts, high intake of plant-based nitrates was associated with a 21% reduction in glaucoma risk [8]. Moreover, some studies suggest that the dysfunction of the NO-Guanylate Cyclase (NO-GC) pathway is associated with glaucoma incidence [52,53]. Blood flow regulation is likely mediated by NO and NO-GC-cyclic guanosine monophosphate [52], and NO also has well known anti-inflammatory and anti-apoptotic properties [54]. The MD is rich in polyphenols, and these components increase the endothelial NO synthase expression, secondarily increasing plasma NO levels [55,56]. The Mediterranean diet also contains many foods rich in L-arginine and nitrate, two key substrates for endogenous NO generation [57]. Previous studies have been published regarding the protective effect on glaucoma incidence of different oral supplements of specific nutrients, such as omega 3, antioxidant-vitamins, or coenzyme q10 [7,58,59]. The Mediterranean diet includes a high consumption of olive oil and oily-fish (rich in omega-3), fruits and vegetables (rich in vitamins), and seeds and nuts (rich in coenzyme q10). Therefore, the MD is a natural source of anti-inflammatory nutrients.

The intimate mechanism of the protective effect of the ML on glaucoma is still not well defined, and it is probably a combined mechanism. However, the ML decreases chronic inflammation markers and endothelial dysfunction parameters while increasing NO plasma levels, and these changes in biological markers might be the cause of the protective effect of the ML on glaucoma incidence. More specialized studies are needed to better understand the intrinsic and complex mechanism of the association between the ML and glaucoma.

The current study has some strengths. It is a prospective study, performed in a large cohort followed up for over 10 years. A wide variety of potential confounders have been taken into account avoiding the possibility of inverse causation bias, which is a frequent phenomenon in cross-sectional or case–control studies. Additionally, the self-reported diagnosis of glaucoma was previously validated in a subsample.

This study also has some limitations that need to be addressed. It is possible that some participants might have overestimated or exaggerated their healthy habits. However, if there was some degree of misclassification, it could be expected to be non-differential, as the bias introduced would tend toward the null hypothesis, therefore not affecting the association found. The lifestyle information was collected on the baseline questionnaire, and although dietary habits and other lifestyle habits may change over time, they are usually stable and reflect both the characteristics of an individual person and those of a type of culture [20,60]. It is known that many patients with glaucoma are unaware of their disease, so it is possible that there are some undiagnosed cases among our participants; this fact probably underestimates the numbers reported in this study. However, since it is unlikely that glaucoma-underdiagnosis is associated with other variables in the study, the most likely misclassification is nondifferential, which, in any case, will bias the results towards the null value, therefore not affecting the association found. In addition, this study was performed in Spain, mostly including only the Caucasian population, therefore, the results should be validated in other ethnic groups or races. Finally, this study was carried out using self-reported information, but self-reported questionnaires have been commonly used in other large cohorts to analyse the relationship between lifestyle and different outcomes. The SUN questionnaires used in this study have been previously validated [16], and have demonstrated efficacy in determining the relationship between lifestyle and several health outcomes, such as metabolic syndrome or cardiovascular disease [12,13]. Despite these limitations, this study is robust in terms of the number of participants and the follow-up time. This study is the first to find that a healthy lifestyle score can decrease the incidence of glaucoma, providing a new modifiable risk factor to prevent the glaucoma disease.

## 5. Conclusions

Our study shows that a SHLS > 6 is a protective factor for glaucoma incidence, and that for each point added to this healthy lifestyle score, the risk of glaucoma decreases by 5%. Thus, the ML is a potentially modifiable factor in the incidence of glaucoma. Further studies are required to validate our results in other cohorts and to better understand the cellular mechanisms behind the association between the Mediterranean lifestyle and the incidence of glaucoma.

## Figures and Tables

**Figure 1 nutrients-14-00779-f001:**
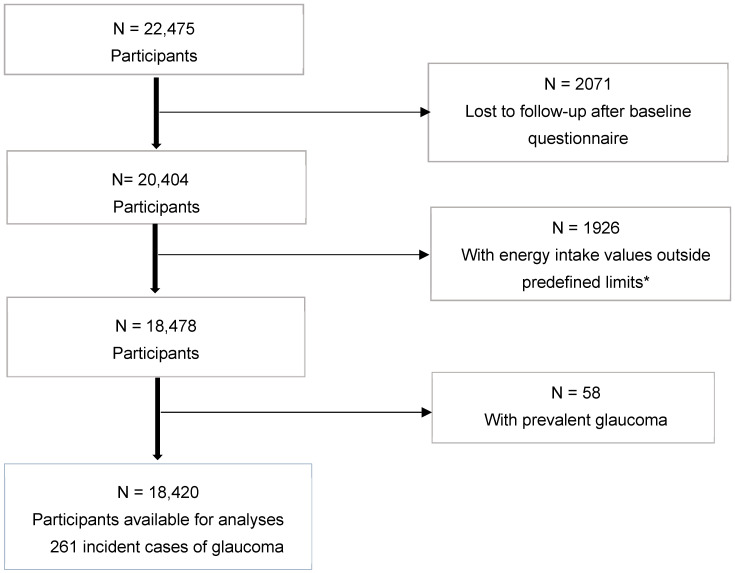
Flow-chart of the subjects in the Seguimiento Universidad de Navarra (SUN) Project, 12 years of follow-up. * Values out of predefined limits according to Willett [15].

**Figure 2 nutrients-14-00779-f002:**
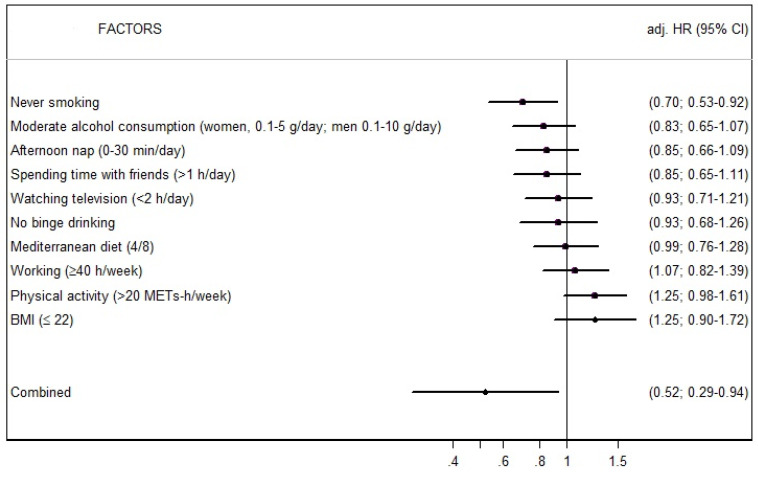
Risk of glaucoma incidence for each factor of the SUN Healthy Lifestyle Score. (The SUN Project). Adjusted for age, sex, calorie intake, caffeine intake, alcohol intake, omega-3/omega-6 ratio, prevalence of cancer, prevalence of hypertension, prevalence of diabetes mellitus type 2, educational level, and following a special diet.

**Table 1 nutrients-14-00779-t001:** Baseline characteristics of participants according to number of healthy lifestyle factors. (The SUN Project).

Number of Healthy Lifestyle Factors	0–2	3–4	5–6	7–10
Participants, *n*	720	5048	8440	4212
Sex, women (%)	49.3	55.0	59.4	69.9
Age, years	41.4 (13)	39.9 (12.7)	37.8 (12.1)	34.3 (10.8)
Body mass index, kg/m^2^	25.5 (3.6)	24.6 (3.5)	23.5 (3.5)	22.1 (3.0)
Alcohol consumption, g/day	12.6 (19)	8.5 (12.5)	6.2 (8.8)	4.2 (5.7)
Smoking, pack year	13.5 (14)	9.2 (11.5)	5.5 (9.3)	2.1 (5.9)
Physical activity, MET-h/week	12.1 (12.3)	16.9 (17.4)	23 (21.9)	32 (25.3)
Mediterranean diet pattern	3.4 (1.4)	3.9 (1.7)	4.2 (1.8)	4.7 (1.7)
Watching television, h/day	2.6 (1.4)	2 (1.4)	1.5 (1.2)	1.1 (0.9)
No binge drinking (%)	35.3	55.9	72.3	85.9
Afternoon nap, min/day	0.2 (0.4)	0.4 (0.5)	0.6 (0.5)	0.8 (0.4)
Meeting up with friends, h/day	0.9 (0.9)	1.2 (1.0)	1.3 (1.0)	1.6 (1.0)
Working ≥40 h/week (%)	17.5	38.3	54.5	70.8
Prevalent diabetes (%)	2.6	2.0	2.0	1.3
Prevalent hypertension (%)	17.2	14.3	10	6.1
Prevalent cancer (%)	2.9	4.3	3.7	3.3
Educational level				
No college (%)	8.9	8.4	9.5	10.4
College (%)	24.2	24.8	23.4	23.2
Post-graduate (%)	52.9	50.1	48.4	46.7
Master (%)	6.7	6.8	8.2	9.4
Doctorate (%)	7.4	9.9	10.4	10.2
Caffeine, mg/day	49.6 (43.2)	44.1 (41.3)	42.7 (39.6)	38.6 (36.9)
Diet (%)	9.0	8.8	8.0	8.4
Total caloric intake	2271.5 (629.5)	2295.1 (625.1)	2344 (614.1)	2411.7 (604.1)

Values are expressed as mean (standard deviation) unless otherwise noted.

**Table 2 nutrients-14-00779-t002:** Glaucoma risk at 10-year follow-up, according to number of healthy lifestyle factors. (The SUN Project).

	Number of Healthy Lifestyle Factors				
	0–2	3–4	5–6	7–10	*p* for Trend
Participants, *n*	720	5048	8440	4212	
Incident cases	18	76	135	32	
Crude	1 (ref.)	0.63 (0.38–1.06)	0.81 (0.50–1.33)	0.53 (0.30–0.96)	0.032
Adjusted for age and sex	1 (ref.)	0.66 (0.39–1.11)	0.84 (0.51–1.37)	0.54 (0.30–0.97)	0.022
Multivariable adjustment *	1 (ref.)	0.65 (0.39–1.10)	0.82 (0.50–1.35)	0.52 (0.29–0.94)	0.15

* Adjusted for age, sex, calorie intake, caffeine intake, alcohol intake, omega-3/omega-6 ratio, prevalence of cancer, prevalence of hypertension, prevalence of diabetes mellitus type 2, educational level, and following a special diet.

**Table 3 nutrients-14-00779-t003:** Contribution of each point to the total variability in the Healthylife Score. (SUN project).

Healthylife Score	% Contribution of Point
No binge drinking	13.7%
Watching television (<2 h/day)	13.4%
Spending time with friends (>1 h/day)	11.7%
Afternoon nap (0–30 min/day)	10.7%
Working (≥40 h/week)	10.0%
Moderate alcohol consumption (women, 0.1–5.0 g/day; men 0.1–10.0 g/day)	9.3%
Never smoking	8.6%
Mediterranean diet (4/8)	8.2%
Physical activity (>20 METs-h/week)	7.8%
BMI (≤22)	6.6%

## Data Availability

The principal investigators, Javier Moreno-Montañés and Alejandro Fernández-Montero have full access to all of the study data and are responsible for the integrity of the data and the accuracy of the data analysis.

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
