# Peer review of "Healthy Lifestyle Score and Incidence of Glaucoma: The Sun Project"

_nutrients, 2022, doi:10.3390/nu14040779_

Round 1

Reviewer 1 Report

The manuscript entitled "HEALTHY LIFESTYLE SCORE AND INCIDENCE OF GLAUCOMA: THE SUN PROJECT" is based on new modifiable risk factors and potential lifestyle changes useful to reduce glaucoma incidence. The role of diet and healthy lifestyle are assessed in glaucoma incidence. The topic is of clinical interest and can help promote a healthy lifestyle in ophthalmology patents from a scientific point of view. There are numerous studies regarding healthy lifestyle and diet for age related macular degeneration, yet very few regarding glaucoma.

The study is nicely presented and based on a large cohort of patients during a 12-year follow-up, thus quite robust numerically speaking. There are, however, a few limitations regarding the study. The incidence of glaucoma reported was 1.42%, which is rather low compared to percentages reported in current literature that tend to be about 1.9% that tends to be greater in cohorts involving older patients. The analysis is based on self-reported diagnosis, which does not give a true picture of disease incidence, especially in glaucoma that tend not to have early visual symptoms, unlike maculopathies or other ocular pathologies. It is not clear how many patients in the cohort underwent a complete ophthalmologic examination within 12 months of self-reporting, which may explain the low numbers. Moreover, glaucoma diagnosis needs to be based on accurate visual field results with repeated and confirmed reduced visual field sensitivity assessed by a glaucoma specialist or valid reading center. These limitations must be reported, which could explain the probable underestimated numbers reported in this study.

On page 9, the authors state the factors that could be related to higher risk of glaucoma based on chronic low-grade inflammation, nitric oxide, altered blood flow and endothelial disfunction. This paragraph is of key importance in explaining the indirect mechanisms involved in a protective factor for glaucoma in patients with a SHLS>6. The authors should provide additional information and more pertinent current studies to better develop this section.

The authors discuss the importance of a healthy diet. Mention should be made regarding studies based on oral supplements and nutrients in the field of glaucoma, like omega 3, Ginkgo biloba, vitamins, coenzyme q10, etc. The authors should comment on how these nutrients can be favored and naturally found in the Mediterranean lifestyle, in comparison to other types of diets.

One of the protective factors considered included working at least 40h/week. This is rather debatable and seems to be promoting a lifestyle totally opposite to the other protective factors of a relaxing lifestyle filled with short napping, meeting with friends and outdoor activities. This factor is confounding, and should be better defined, especially considering that the average Mediterranean and public servants work a maximum of 38 hours/week. This factor needs either to be removed or redefined with a maximum (for example working 35-45 hours/week).

Author Response

Dear Editor,

We provide below a point-by-point answer to all the issues and queries raised by the reviewers, in red font. Corrections are also highlighted in the original manuscript in red font.

We deeply appreciate the review performed, which will certainly increase the quality of the manuscript. A new version approved by all authors has been already uploaded to the submission platform. We hope our answers and the changes performed to the manuscript are satisfactory; but please let us know if there is any other correction needed.

All authors have approved the last version of the manuscript.

Comments and Suggestions for Authors

The manuscript entitled "HEALTHY LIFESTYLE SCORE AND INCIDENCE OF GLAUCOMA: THE SUN PROJECT" is based on new modifiable risk factors and potential lifestyle changes useful to reduce glaucoma incidence. The role of diet and healthy lifestyle are assessed in glaucoma incidence. The topic is of clinical interest and can help promote a healthy lifestyle in ophthalmology patents from a scientific point of view. There are numerous studies regarding healthy lifestyle and diet for age related macular degeneration, yet very few regarding glaucoma.

The study is nicely presented and based on a large cohort of patients during a 12-year follow-up, thus quite robust numerically speaking.

A.- Thanks for your comment.

Q.-  There are, however, a few limitations regarding the study. The incidence of glaucoma reported was 1.42%, which is rather low compared to percentages reported in current literature that tend to be about 1.9% that tends to be greater in cohorts involving older patients. The analysis is based on self-reported diagnosis, which does not give a true picture of disease incidence, especially in glaucoma that tend not to have early visual symptoms, unlike maculopathies or other ocular pathologies. It is not clear how many patients in the cohort underwent a complete ophthalmologic examination within 12 months of self-reporting, which may explain the low numbers. Moreover, glaucoma diagnosis needs to be based on accurate visual field results with repeated and confirmed reduced visual field sensitivity assessed by a glaucoma specialist or valid reading center. These limitations must be reported, which could explain the probable underestimated numbers reported in this study.

A.- It is true that the incidence of glaucoma in our participants is lower than the incidence reported in current literature. This might be due to several factors, On one hand, glaucoma is an age-related disease and the mean age of our participants is around 40 years, also the questionnaires did not specifically collect if participants had been to an ophthalmology department for examination in the previous months. These facts may be a study limitation. However, the aim of the study is not to estimate glaucoma incidence in our participants, but to analyze modifiable glaucoma risk-factors.

Glaucoma was self-reported, however, most of our participants are university graduates (many from medicine or nursing) and we believe that their answers are valid. Finally, the validation study with 150 participants showed almost perfect agreement, which reinforces the validity of our results. Following your suggestions, we have added in the limitations paragraph the following sentence:

" It is known that many patients with glaucoma are unaware of their disease, so it is possible that there are some undiagnosed cases among our participants, as they were not all seen by an ophthalmologist; this fact probably underestimates the numbers reported in this study. However, since it is unlikely that glaucoma-underdiagnosis is associated with other variables in the study the most likely misclassification is non differential, which, in any case, will bias the results towards the null value, therefore not affecting the association found.”

Q.- On page 9, the authors state the factors that could be related to higher risk of glaucoma based on chronic low-grade inflammation, nitric oxide, altered blood flow and endothelial disfunction. This paragraph is of key importance in explaining the indirect mechanisms involved in a protective factor for glaucoma in patients with a SHLS>6. The authors should provide additional information and more pertinent current studies to better develop this section.

A.- We have improved the discussion, adding different possible pathways by which the Mediterranean diet may protect against glaucoma, especially through the nitric oxide pathway. We have also increased the number of references including additional recent articles.

Q.- The authors discuss the importance of a healthy diet. Mention should be made regarding studies based on oral supplements and nutrients in the field of glaucoma, like omega 3, Ginkgo biloba, vitamins, coenzyme q10, etc. The authors should comment on how these nutrients can be favored and naturally found in the Mediterranean lifestyle, in comparison to other types of diets.

A.-Thanks for your comment, we totally agree, and the following sentence has been added to the reviewed version of the manuscript:

“Previous studies have been published regarding the protective effect on glaucoma incidence of different oral supplements of specific nutrients, such as omega 3, antioxidant-vitamins or coenzyme q10. Mediterranean diet includes a high consumption of olive oil and oily-fish (rich in omega-3), fruits and vegetables (rich in vitamins) seeds and nuts (rich in coenzyme q10) therefore it is a natural source of these anti-inflammatory nutrients”.   

Q.- One of the protective factors considered included working at least 40h/week. This is rather debatable and seems to be promoting a lifestyle totally opposite to the other protective factors of a relaxing lifestyle filled with short napping, meeting with friends and outdoor activities. This factor is confounding, and should be better defined, especially considering that the average Mediterranean and public servants work a maximum of 38 hours/week. This factor needs either to be removed or redefined with a maximum (for example working 35-45 hours/week).

A.-Thanks for your comment. We understand your concerns regarding this factor, but it is part of a previously created score. The SUN Healthy Lifestyle Score (SHLS) includes 10 habits. We have made previous studies in our cohort using this same score, and in other previous published studies from the SUN cohort regarding metabolic syndrome, cardiovascular disease or all-cause mortality, the SHLS has proven to be a protective factor. We understand the reviewer concerns, regarding this item, but it is a part of a previously defined score, and therefore it cannot be changed nor excluded, if we want to use the “SUN Healthy Lifestyle Score”. Nevertheless, following the reviewer suggestion we have performed some sensibility analysis, excluding the “working factor”  from our SHLS index and the results obtained remain consistent (HR: 0.52 (0.29 – 0.94) when using the SHLS, and HR: 0.51 (0.26-0.99), when excluding the factor related to working hours.

Reviewer 2 Report

This is a very well written study that leads to the conclusion, that  a healthy lifestyle score can decrease the incidence of  glaucoma, providing a new modifiable risk factor to prevent the glaucoma disease. The authors have designed the score, which includes 10 items and when they were analyzed together all of them, they showed a significant protective effect in glaucoma.

The authors discuss the biological pathway behind the association between Mediterranean Lifestyle (ML) and the reduction of glaucoma new cases. They suppose a role of low-grade inflammation and decrease of the endothelial  nitric oxide (NO) synthesis that may be improved in  ML. We recommend to cite the first study on NO in glaucoma pathogenesis published in 2001 (Nitrogen oxide in the pathogenesis of glaucoma and cataract  Vestn.Oftalmol. Sep-Oct 2001;117(5):34-7. PMID: 11765466).

The only weak side of the study is that it was conducted in Spain, therefore, its results cannot be extrapolated to other ethnic groups. All identified cases of glaucoma were related to POAG. Meanwhile, in a number of regions of the globe, other forms are common, for example, pseudo-exfoliative. It is not clear whether the authors' recommendations can apply to these forms of glaucoma. This should be noted as a limitation of the study.

Author Response

Dear Reviewer,

We provide below a point-by-point answer to all the issues and queries raised. Corrections are also highlighted in the original manuscript in red font.

We deeply appreciate the review performed, which will certainly increase the quality of the manuscript. A new version approved by all authors has been already uploaded to the submission platform. We hope our answers and the changes performed to the manuscript are satisfactory; but please let us know if there is any other correction needed.

All authors have approved the last version of the manuscript.

Comments and Suggestions for Authors

Q.- This is a very well written study that leads to the conclusion, that  a healthy lifestyle score can decrease the incidence of  glaucoma, providing a new modifiable risk factor to prevent the glaucoma disease. The authors have designed the score, which includes 10 items and when they were analyzed together all of them, they showed a significant protective effect in glaucoma.

A.- Thanks for your comments.

Q.- The authors discuss the biological pathway behind the association between Mediterranean Lifestyle (ML) and the reduction of glaucoma new cases. They suppose a role of low-grade inflammation and decrease of the endothelial  nitric oxide (NO) synthesis that may be improved in  ML. We recommend to cite the first study on NO in glaucoma pathogenesis published in 2001 (Nitrogen oxide in the pathogenesis of glaucoma and cataract  Vestn.Oftalmol. Sep-Oct 2001;117(5):34-7. PMID: 11765466).

A.- Thanks for your suggestion, The article has been included in the references. We have also improved the discussion adding new information on the mechanisms by which MD protects from glaucoma through the NO pathway.

Q.- The only weak side of the study is that it was conducted in Spain, therefore, its results cannot be extrapolated to other ethnic groups. All identified cases of glaucoma were related to POAG. Meanwhile, in a number of regions of the globe, other forms are common, for example, pseudo-exfoliative. It is not clear whether the authors' recommendations can apply to these forms of glaucoma. This should be noted as a limitation of the study.

A.-Yes, we agree with you. This limitation has been addressed and added in the limitations section. This study has been performed in Spain, mostly including Caucasian population, therefore, the results should be validated in other ethnic groups or races.

Round 2

Reviewer 1 Report

The authos addessed all issues raised in the review.